



# Modelling changes in secondary inorganic aerosol formation and nitrogen deposition in Europe from 2005 to 2030

Jan Eiof Jonson[1], Hilde Fagerli[1], Thomas Scheuschner[2], and Svetlana Tsyro[1]

[1]Norwegian Meteorological Institute, Oslo, Norway
[2]Umweltbundesamt, Dessau-Roßlau Germany

**Correspondence:** Jan Eiof Jonson (j.e.jonson@met.no)

**Abstract.**

Secondary inorganic $PM_{2.5}$ particles are formed from $SO_x$, $NO_x$ and ammonia emissions, through the formation of either ammonium sulphate or ammonium nitrate. EU limits and WHO guidelines for $PM_{2.5}$ levels are frequently exceeded in Europe, in particular in the winter months. In addition the critical loads for eutrophication are exceeded in most of the European

continent. Further reductions in ammonia emissions and other PM precursors beyond the 2030 requirements could alleviate some of the health burden from fine particles, and also reduce the deposition of nitrogen to vulnerable ecosystems.

Using the regional scale EMEP/MSC-W model, we have studied the effects of year 2030 ammonia emissions on $PM_{2.5}$ concentrations and depositions of nitrogen in Europe in the light of present (2017) and past (2005) conditions. Our calculations show that in Europe the formation of $PM_{2.5}$ from ammonia to a large extent is limited by the ratio between the emissions of

10 ammonia on one hand, and $SO_x$ plus $NO_x$, on the other hand. As the ratio of ammonia to $SO_x$ and $NO_x$ is increasing, the potential to further curb $PM_{2.5}$ levels through reductions in ammonia emissions is decreasing. Here we show that per gram of ammonia emissions mitigated, the resulting reductions in $PM_{2.5}$ levels simulated using 2030 emissions are about a factor of 2.6 lower than when 2005 emissions are used. However, this ratio is lower in winter, thus further reductions in the ammonia emissions in winter may have similar potentials as $SO_x$ and $NO_x$ in curbing $PM_{2.5}$ levels in this season.

Following the expected reductions of ammonia emission, depositions of reduced nitrogen should also decrease in Europe. However, as the reductions in $NO_x$ emission are larger than for ammonia, the fraction of total nitrogen (reduced plus oxidised nitrogen) deposited as reduced nitrogen is increasing and may exceed 60% in most of Europe by 2030. Thus the potential for future reductions in the exceedances of critical loads for eutrophication in Europe will mainly rely on the ability to reduce ammonia emissions.



## 1 Introduction

Concentrations of particles with a diameter of less than 2.5 $\mu$m ($PM_{2.5}$) have been decreasing in most of Europe since the turn of the century as a combined result of reductions in anthropogenic emissions of primary particles and gaseous $PM_{2.5}$ precursors. Emissions of ammonia play a central role in the secondary particle formation, and are also major contributors to

25 the exceedances of critical loads for eutrophication (Tsyro et al., 2020). In most parts of Europe emissions of in particular $SO_x$ (emitted predominantly as $SO_2$ but also as $SO_4^{2-}$, (hereafter $SO_4$) and $NO_x$ ($NO + NO_2$) have been steadily decreasing in the past decades. At the same time, emissions of ammonia have changed much less, decreasing in some European countries and increasing in others (see EMEP Status Report 1/2020 (2020), chapter 3 and Appendix B). Further reductions of $SO_x$, $NO_x$, and ammonia emissions are required by the year 2030 according to the EU NEC2030 directive (https://www.eea.europa.eu/

themes/air/air-pollution-sources-1/national-emission-ceilings), but the projected percentage reductions in ammonia emissions in NEC2030 are smaller than for $SO_x$ and $NO_x$. In the atmosphere $SO_2$ is oxidised to $SO_4$ and $NO_x$ to $HNO_3$. Contrary to $SO_x$ and $NO_x$, more than 90% of the ammonia emissions are from agriculture, with only minor contributions from industry and traffic (IIASA, 2020). As a result these emissions are in general not co-located with the $SO_x$ and $NO_x$ emissions. In addition the temporal distribution of the emissions differ, with ammonia emissions peaking in spring and summer, whereas

anthropogenic $SO_x$ and $NO_x$ emissions in general peak in winter. In the fine mode, ammonium sulphate particles are first formed from ammonia and $H_2SO_4$. Any excess ammonia can then form ammonium nitrate ($NH_4NO_3$) in thermodynamic equilibrium with $HNO_3$ (see e.g. Simpson et al. (2012)). When ammonia is in excess relative to both $H_2SO_4$ concentrations and the equilibrium with $HNO_3$, the formation of ammonium salts will slow down at some point (when there is less acid available to react with ammonia), and free ammonia will be present. With ammonia emissions greatly exceeding $SO_x$ and

$NO_x$ emissions already before 2005, one could question the effects of small or moderate reductions in ammonia emissions on Secondary Inorganic Aerosols contributing to $PM_{2.5}$ (SIA25), a major component in $PM_{2.5}$.

Using emissions as described in Jiang et al. (2020), Aksoyoglu et al. (2020) showed that the fraction of ammonium in SIA25 was similar when calculated with 1990 versus 2030 emissions. With 1990 versus 2030 emissions the fraction of sulphate in SIA25 dropped significantly, whereas the nitrate fraction increased, compensating for the reduction in sulphate In many air

pollution episodes in Europe involving $PM_{2.5}$, ammonium nitrate has accounted for a large portion of the aerosol mass (Petit et al., 2017; Vieno et al., 2016). With a large surplus of $NH_3$ relative to $HNO_3$ it could be that ammonium nitrate formation will be virtually unaffected by changes in ammonia emissions. Both $NO_x$ and ammonia are relatively shortlived, with a lifetime in the atmosphere of about 1 day (Seinfeld and Pandis, 2016). Given the difference in both spatial and temporal distribution in the sources of ammonia, $NO_x$ and $SO_x$, substantial local variability in the ratio between ammonia on one hand, and sulphate

and/or $HNO_3$ on the other hand, can be expected. Thus, locally the formation of SIA25 may be limited by the availability of either ammonia or by $HNO_3$ and sulphat due to the lack of co-location of the sources of these species.

Here we apply the EMEP MSC-W model to investigate how $PM_{2.5}$ concentrations, and deposition of reduced nitrogen, have changed from 2005 to 2017. But the main focus is on model calculations for 2030, assuming that the NEC2030 requirements will be met. Given that ammonia concentrations in Europe are generally in substantial excess of $HNO_3$ concentrations, we





explore to what extent additional reductions in ammonia emissions will contribute to further reductions in ammonium and
subsequently to reductions in $PM_{2.5}$ levels, and to what extent the response to further ammonia emissions is linear. We try to
answer this with a sensitivity study for $PM_{2.5}$ for post NEC2030, applying step-wise additional ammonia emission reductions
on top of the NEC2030 requirements holding all other emissions constant. At the same time we also investigate to what extent
reductions in ammonia emissions may affect deposition of reduced nitrogen and the exceedance of critical loads for nitrogen
deposition.

## 2   Model description

The model calculations have been made with the EMEP MSC-W model (hereafter 'EMEP model'), version rv4.34, on 0.1 x 0.1
° resolution for the domain between 30° W and 45° E and between 30 and 75° N. A detailed description of the EMEP model
can be found in Simpson et al. (2012), with later model updates described in Simpson et al. (2020) and references therein. In
the EMEP model the composition of the metastable aqueous aerosols of the inorganic system $NO_3^- - NH_4^+$ and water, and $NH_3$
– $HNO_3$ in the gas phase, are calculated using the MARS equilibrium model (Binkowski and Shankar, 1995). In Tsyro and
Metzgert (2019) the EMEP model results using the MARS model are compared to model calculations with EQUSAM4clim
(Metzger et al., 2016, 2018) giving very similar results.

The EMEP model is available as open source code (see code availability) and is under continuous development receiving
feedback from a host of users. It is regularly evaluated against measurements, see Gauss et al. (2017, 2018, 2019, 2020) for the
most recent evaluations. Scatter plots of model versus measurements for the concentrations of several key species as well as
for the wet depositions of reduced and oxidised nitrogen are shown in appendix A. The model performance is comparable for
both 2005 and 2017, even though the selection of measurement sites differ for the two years. Measurements are available for
subsets of common sites for the two years, in general showing comparable model to measurement biases for the concentrations
of key species for the years 2005 and 2017. The EMEP model has also participated in model intercomparisons and model
evaluations in a number of peer reviewed publications (Karl et al., 2019; Colette et al., 2011, 2012; Jonson et al., 2018). In
Vivanco et al. (2018) depositions of sulphur and nitrogen species in Europe have been calculated by 14 regional models and
compared to measurements, and in Theobald et al. (2018) the model calculated the trends in the wet deposition of sulphur as
well as reduced and oxidised nitrogen from 6 models, including the EMEP model, are compared to measurements from 1990
to 2010. Both these two studies showed good results for the EMEP model.

### 2.1   Definition of the Critical Load for eutrophication

A Critical Load (CL) is defined as "a quantitative estimate of an exposure to one or more pollutants below which significant
harmful effects on specified sensitive elements of the environment do not occur according to present knowledge" (Nilsson and
Grennfelt, 1988). CLs are calculated for different receptors (e.g. terrestrial ecosystems, aquatic ecosystems), and a sensitive
element can be any part (or the whole) of an ecosystem or ecosystem process. CLs have been derived for several pollutants
and different negative effects. Here we restrict ourselves to CL defined to avoid the eutrophying effects of Nitrogen deposition





(CLeutN). Like sulphur, nitrogen can also cause acidifying impacts in ecosystems, but the areas affected by acidification are strongly decreasing in Europe compared to earlier decades, and therefore the focus of this paper is on the eutrophying effects (Slootweg et al., 2015; EEA, 2014; Hettelingh et al., 2017).

The CLeutN for a site is either empirically derived or calculated from steady-state simple mass balance (SMB) equations. In the SMB method, non-harmful nitrogen-fixing processes are described mathematically and combined with a chemical criterion (e.g., an acceptable N concentration in the soil solution). This summation is then compared to the corresponding deposition value. Methods to compute CLs are summarised in the Mapping Manual of the ICP Modelling and Mapping CLRTAP (2017), (see also De Vries et al. (2015)), which is used within the Convention on Long-range Transboundary Air Pollution: https:

95 //unece.org/40-years-clean-air.

    If the deposition of the pollutant under consideration is greater than the CL at a site, the CL is designated as exceeded. Such site-specific exceedances can be summarised for different spatial entities (e.g. grid cells, countries). This method is called average accumulated exceedance, and is defined as the weighted average of exceedances for all ecosystems within the selected region, where the weights are the respective ecosystem areas (Posch et al., 2001).

The CL exceedances presented here were calculated using the current CL database, which is described in Hettelingh et al. (2017) and stored by the current Coordination Centre for Effects (CCE) at the German Federal Environmental Agency. This dataset is also used, among other things, to support European assessments and negotiations on emission reductions (Hettelingh et al., 2001; Reis et al., 2012; EEA, 2014).

## 3   Model runs

The EMEP model runs have been performed with 2017 meteorological conditions. In these model runs, emissions estimated for the years 2005, 2017, and projected for 2030, have been used. For the EU28 countries, the official EMEP emissions have been used for both the 2005 and 2017 model runs (as listed in EMEP Status Report 1/2020 (2020), appendix B). For the 2030 model runs the emissions for the individual EU28 countries are scaled from the 2005 emissions according to the NEC2030 obligations. The total emissions of ammonia, $SO_x$ and $NO_x$ in the EU28 countries in 2005, 2017 and 2030 are illustrated in

Figure 1a. For all other countries and regions the 2005 and 2030 emissions have been provided by the International Institute for Applied Systems Analysis (IIASA) within the European FP7 project ECLIPSE (http://www.iiasa.ac.at/web/home/research/researchPrograms/air/ECLIPSEv5.html). In this study we use ECLIPSE version 6a (hereafter referred to as 'ECLIPSEv6a'), which is a global emission data-set widely used by the scientific community. Some of the methods used in ECLIPSEv6a are described in the recent publication of Höglund-Isaksson et al. (2020). Ammonia emissions from all EU28 countries and

selected European non-EU countries are listed in Table 1.

    In order to explore the effects that further emission reductions of ammonia in 2030 may have on $PM_{2.5}$ concentrations and nitrogen depositions, additional model sensitivity runs have been made. 2030 ammonia emissions have been reduced by up to 50% in steps of 10%. In addition the 2030 emissions of $SO_x$ and $NO_x$ have been reduced by 10%, and the 2005 ammonia emissions by 10%. All model runs are listed in Table 2.



## 4 Model results for 2005 versus 2030

### 4.1 PM$_{2.5}$

Figure 2 upper panels, shows concentrations of PM$_{2.5}$ as calculated with 2005 emissions (upper left) and 2030 emissions (upper right). The high PM$_{2.5}$ levels over North Africa in both 2005 and in 2030 are caused by large natural sources of mineral dust. As shown in Figure 2, substantial reductions in PM$_{2.5}$ concentrations are expected from 2005 to 2030, caused by reductions in ammonia emissions, and even larger reductions in SO$_x$ and NO$_x$ emissions, in Europe. Even so, in 2030 elevated PM$_{2.5}$ concentrations still persist in some areas, notably in the Po Valley in Italy and the BeNeLux countries (Belgium, The Netherlands and Luxembourg). In these areas, anthropogenic primary PM$_{2.5}$ and PM$_{2.5}$ precursor emissions are expected to remain high also in 2030. As a result the limit values for PM$_{2.5}$ recommended by WHO (WHO, 2005) are expected to be exceeded in these locations also in 2030.

Figure 1b shows the concentrations of SIA25 averaged over the EU28 countries in 2005, 2017, and 2030 split by component (sulphate, nitrate, and ammonia). Even if the reductions in ammonia emissions in EU28 are much smaller than the corresponding reductions in SO$_x$ and NO$_x$, the calculated percentage contributions to SIA25 from ammonium are virtually unchanged between 2005, 2017, and 2030, confirming the findings in Aksoyoglu et al. (2020) for the Payern measurement site in Switzerland. The lack of change in the fraction of ammonium is not surprising, as it is either associated with sulphate ($(NH_4)_2SO_4$) or with nitrate ($NH_4NO_3$). As the molecular weight is 18 g/mol for NH$_4$, 96 g/mol for SO$_4$, and 62 g/mol for NO$_3$, the resulting percentage contribution by weight from NH$_4$ for both ammonium nitrate and ammonium sulphate is roughly 25%, consistent with the contributions shown in Figure 1b. Between 2005 and 2017 the percentage reductions in SO$_x$ emissions in EU28 were more that twice as large as the reductions in NO$_x$, resulting in an increase in the nitrate fraction in SIA25. From 2017 to 2030 the EU28 reductions in NO$_x$ are expected to be larger than for SO$_x$, resulting in a slight decrease in the fraction of nitrate, and a corresponding increase in the sulphate fraction in SIA25.

### 4.2 Deposition of reduced nitrogen

Figure 3 shows the depositions of reduced nitrogen in 2005 (left) and in 2030 (right). As for PM$_{2.5}$, the Po valley and the BeNeLux countries stand out, receiving large amounts of reduced nitrogen depositions both in 2005 and in 2030. The total amount of deposition of reduced nitrogen (and also oxidised nitrogen) per country in 2005, 2017, and 2030 are listed in Table 1. In most central European high emitting countries less reduced nitrogen is deposited than they emit. Several countries facing the sea, with very few upwind sources, exemplified by Ireland and Portugal, receive far less deposition than they emit. At the same time the Nordic countries (Norway, Sweden, and Finland) and the Baltic countries (Estonia, Latvia, and Lithuania), located downwind of Central Europe, receive more depositions of reduced nitrogen than they emit. For the European Union as a whole, the fraction of deposited over emitted reduced nitrogen is between 0.7 and 0.8 for all 3 emission years considered. The remaining 0.2-0.3 is either deposited at sea or in non-EU countries. Only a small portion is advected out of the model domain.

As a result of the lower ambitions for reductions in ammonia emissions compared to NO$_x$ emissions, a larger portion of the total nitrogen deposition is expected to come from ammonia. This is illustrated in Figure 4 which shows the fraction of reduced





nitrogen in the total nitrogen deposition calculated with 2005, 2017, and 2030 emissions. The figure shows that this fraction increases significantly from 2005 to 2017, with a further increase expected from 2017 to 2030. By 2030 the percentage of the
155 total nitrogen deposition resulting from ammonia emissions is expected to exceed 60% in large parts of Europe. The percentage contributions are also listed as an average for EU28, and as averages for European countries in Table 1. This underpins the findings from IIASA (2018), that the potential for further reductions of the exceedances of CL for eutrophication is mainly depending on our ability to control future ammonia emissions. As shown in Figure 5 the calculated CL for eutrophication are exceeded for all three years (2005, 2017, and 2030). Even though the level of exceedance has been substantially reduced from
160 2005 to 2017, and large reductions are expected also from 2017 to 2030, the total area in Europe where the CL is exceeded remains high for all three years. The percentage of the area where the CL for eutrophication are exceeded are listed in Table 3 for individual European countries.

### 4.3   Effects of ammonia emission controls.

Figures 2c and d show the effects of a further 10% reductions of ammonia emissions on $PM_{2.5}$ concentrations in 2005 and
165 2030, respectively. Compared to 2005, the absolute effects of 10% further emission reductions in 2030 are smaller. Partially, this is because the percentage emission reductions in 2005 give a larger reduction in absolute numbers compared to percentage emission reductions based on the lower 2030 emissions. As an example, 10% of the emissions from EU in 2005 (3574Gg) will give a smaller reduction than 10% reductions in 2030 (2900Gg). However, these absolute changes in ammonia emissions are not large enough to explain a decrease of the magnitude seen in Figure 2c versus Figure 2d. As seen in Figure 6, there is
170 more 'free' ammonia (ammonia in excess of $H_2SO_4$ and $HNO_3$) in 2030 relative to 2005. A larger portion of free ammonia could partially explain the 2% - 4% annual increase observed by satellites between 2008 and 2018 in countries like Belgium, the Netherlands, France, Germany, Poland, Italy and Spain (Damme et al., 2020). A 10% reduction in ammonia emissions will make gradually smaller impact on the formation of ammonium in future years. This is exemplified by the EU28 countries in Table 4. For 2005, we find that as an annual average 10% reductions of ammonia emissions were about four times more
efficient than 10% reductions in $NO_X$ and almost twice as efficient as $SO_x$ in reducing $PM_{2.5}$ per Gg emitted. For 2030, we find that as an annual average the efficiency mitigating $PM_{2.5}$ concentrations by reducing ammonia emissions by 10% has been reduced from 0.61 to 0.22 $ngNm^{-3}$ per Gg ammonia emitted, a reduction of a factor of about 2.6 from 2005. Over the same timespan, the efficiency of a further 10% reduction in $NO_x$ emissions has gone up by about a factor of 1.8 (from 0.15 to 0.27) and by about a factor of 1.6 (0.37 to 0.58) for a 10% further reduction in $SO_X$ emissions.

The dry deposition of ammonia is faster than that of ammonium. As the fraction of ammonia in total reduced nitrogen increases from 2005 to 2030 (as discussed in Section 4.1), reduced nitrogen may be deposited closer to its sources and potentially increasingly more in the same country as it is emitted. A trend in deposition versus emissions for the individual countries (deposition divided by emissions in Table 1) is not readily seen based on the model calculations. The geographical extent of the countries in Europe is relatively small, and there is considerable variability in the emission trends for ammonia between
the individual EU28 countries, affecting the trends in the depositions also in neighbouring countries.





As a large portion of the emitted reduced nitrogen is deposited close to its sources, changes in emissions close to the EU28 outer geographical borders should affect this fraction more for EU28 as a whole than emission changes in central parts. Ammonia emissions in large EU28 countries as Germany and France have increased between 2005 and 2017, whereas emissions in several countries close to the eastern and southestern geographical EU28 borders, such as Bulgaria, Romania and
Greece, have decreased. For the EU28 countries as a whole the fraction of deposited over emitted reduced nitrogen is between 0.7 and 0.8 for all three years considered (2005, 2017 and 2030). It would be possible to investigate the hypothesis of a possible decrease of the transport distance of reduced ammonia by looking at so called source receptor matrices for the different years (e.g. studying how the contribution from the country to itself have changed over the years). Such experiments are planned as a follow-up of this paper.

### 4.3.1   Seasonal effects of ammonia emission reductions on $PM_{2.5}$

More than 90% of the ammonia emissions are from agriculture (IIASA, 2020), with low emissions in winter and a maximum in spring, as opposed to both $NO_x$ and $SO_x$ emissions peaking in winter. As a result, there are more sulphate and $HNO_3$ relative to ammonia in winter than in other seasons. Also the condensation process forming ammonium nitrate aerosols is favoured by low temperatures. As a result Figure 7 shows that for $PM_{2.5}$ by far the largest effects of further reductions of
ammonia emissions are modelled for the winter months. Notably, most of $PM_{2.5}$ pollution episodes, including exceedances of the EU limits or WHO AQ guidelines for daily mean $PM_{2.5}$ concentrations, are most frequent in large parts of Europe during the winter period (see Tsyro et al. (2019)). The smallest effects are calculated for the summer months, when both $SO_x$ and $NO_x$ emissions are at a minimum. Furthermore ammonium nitrate is less stable in higher temperatures, and more likely to decompose into gaseous $NH_3$ and $HNO_3$. Thus in summer, reductions are mainly confined to the southwestern parts of
the North Sea, where ship emissions of $NO_x$ are large. This seasonal behaviour is also seen in the measurements at Preila in Lithuania, with low ammonium concentrations in summer and higher concentrations in the cold season (Davuliene et al., 2021). Furthermore, they found that the relative abundance of ammonium nitrate has increased at the expense of ammonium sulphate as a result of particularly large reductions in $SO_x$ emissions in the last decades.

The seasonal behaviour of $PM_{2.5}$ formation from ammonia is also demonstrated for EU28 in Table 4 showing that the $PM_{2.5}$
reductions that can be achieved by reducing ammonia emissions are largest in winter, and are almost constant (and low) for each 10% increment in emission reduction in summer. With a large surplus of free ammonia in summer the impact of further emission reductions are small. In winter the ammonia surplus relative to $HNO_3$ and $SO_4$ is much smaller (or nonexistent), and additional ammonia emission reductions will have larger impacts on $PM_{2.5}$ levels.

### 4.3.2   Sensitivity tests with additional emission controls.

Figure 8 compares the efficiency of ammonia emissions reductions on top of the NEC2030 requirements on $PM_{2.5}$ concentrations and reduced nitrogen depositions. Starting from the expected emission levels in 2030, the maps compare the effects of the first 10% reductions (Base - 10%) in ammonia emissions to the effects of further reductions in ammonia emissions from 40 - 50% relative to Base. If linear, the effects of these 10% increments in emissions should be equal. However, as shown in





Figure 8a, the reductions in $PM_{2.5}$ are larger for the 50%–40% emission reductions compared to 10%–Base reductions almost
everywhere. This is further demonstrated in Table 4, listing the reductions in annual and seasonal $PM_{2.5}$ concentrations as an
average over the EU28 countries in steps of 10% relative to the 2030 NEC emissions. Both as an annual average, and for each
season, the reductions in $PM_{2.5}$ increase for each 10% increment. The reductions in $PM_{2.5}$ increase from 0.23 ngm$^{-3}$ per
Gg ammonia emitted for the first 10% additional reductions, to 0.35 ngm$^{-3}$ per Gg emitted for the 50%–40% reductions. The
increase in efficiency is a result of a shift in the ratio in ammonia versus $SO_x$ and $NO_x$ emissions. In Table 4 we also show
that with the much higher $SO_x$ and $NO_x$ emissions versus ammonia emissions, the potential of 10% additional reductions in
ammonia emissions in curbing $PM_{2.5}$ levels was substantially higher in 2005, even when compared to 40%–50% reductions
in 2030.

In two additional model runs we separately reduce the 2030 emissions of $SO_x$ and $NO_x$ by 10%. As an annual average
we find that 10% reductions in both $SO_x$ and $NO_x$ emissions will lead to larger reductions in $PM_{2.5}$ levels in EU28 than the
corresponding 10% reductions in ammonia emissions. However, as discussed in section 4.3.1, the seasonal variation is large,
and in winter reductions of $PM_{2.5}$ per Gg emitted could still be larger for ammonia than for $NO_x$.

For depositions of reduced nitrogen the situation is reversed. As shown in Figure 8b the reductions in depositions achieved
with ammonia reduced between 50% and 40% compared to the first 10% reductions are (marginally) smaller in the vicinity of
the source regions. This can be explained by a slightly larger portion of the emitted ammonia being converted to ammonium
aerosols having a slower dry deposition rate than ammonia. As a result, the higher deposition seen in the source areas is
compensated by a much smaller, but more widespread decrease elsewhere. Only a small portion of the reduced nitrogen is
advected out of the model domain.

## 5    Conclusions

Focusing on the effects of ammonia emissions we have investigated how $PM_{2.5}$ concentrations and depositions of reduced
nitrogen will change from 2005 to 2030, assuming that the NEC2030 emission targets will be met. In addition, we have made a
sensitivity study for $PM_{2.5}$ for post NEC2030, assuming additional emission reductions on top of the NEC2030 requirements.

Emissions of $SO_x$ and $NO_x$ have decreased in Europe from year 2005 to present, and further emissions reductions are
expected by year 2030. However, ammonia emissions have so far remained high, and projected NEC2030 emission reductions
of ammonia are much smaller than for $SO_x$ and $NO_x$. Our model calculations show that these differences in emission trend lead
to a smaller fraction of the emitted ammonia being converted to ammonium and an increasingly larger portion of free ammonia
versus ammonium in the atmosphere in Europe. Based on 10% emission reductions of $NH_3$, $NO_x$, and $SO_x$ we calculate that
the potential for $PM_{2.5}$ formation per Gg $NH_3$ emitted, is expected to drop by a factor of about 2.6 as an annual average
between 2005 and 2030. Over the same timespan the potential for forming $PM_{2.5}$ from $NO_x$ per Gg emitted has increased by
a factor of 1.8, and from $SO_x$ by a factor of 1.6 per Gg emitted.

In winter, with low ammonia emissions and relatively higher $NO_x$ and $SO_x$ emissions, the ratio between $NH_3$ versus $HNO_3$
and $SO_4$ is higher, and a larger portion of the emitted ammonia will form particulate ammonium. Also, the formation of





ammonium nitrate in equilibrium with $HNO_3$ and $NH_3$ is favoured by low temperatures. As a result we find that in winter the effects of further reductions in ammonia emissions are larger than in other seasons, and comparable to additional reductions in $SO_x$ and $NO_x$ emissions. This is in agreement with the findings in Backes et al. (2016), pointing out that even though the
255 ammonia emissions are highest in spring and summer due to the application of manure on the fields, emission reductions in winter have a stronger impact on the formation of secondary aerosols than in any other season. Furthermore they stated that the potential of reducing ammonia emissions in winter is highest through the reduction of animal farming, as this source accounts for about 80% of the ammonia emissions in the autumn and winter months.

Following the emissions reduction of ammonia, deposition of reduced nitrogen is decreasing in Europe. However, the reduc-
260 tions in $NO_x$ emissions are much larger than for ammonia, resulting in a much faster decline in oxidised nitrogen deposition compared to reduced nitrogen. Thus the fraction of reduced over total deposition of nitrogen is increasing, and is expected to reach more than 60% in large parts of Europe by year 2030. The calculations show that with the existing emission projections the CL for nitrogen will be exceeded in large parts of Europe also in 2030. There are also indications that reduced nitrogen inputs are more effective in decreasing biodiversity than is oxidised nitrogen, as reduced nitrogen is more readily available,
stimulating growth of specific plants at the expense others (see van den Berg et al. (2008) and Erisman et al. (2007) and references therein). Furthermore they also suggest that increased levels of ammonium can be toxic to plants (see also Esteban et al. (2016)).

Reducing, and preferably removing, these exceedances, will require larger reduction in nitrogen emissions than currently projected. Given that reduced nitrogen is responsible for the major fraction of nitrogen depositions, the largest cuts should be
made in the ammonia emissions.

For many countries the latest source oriented legislation may potentially reduce the emissions of $SO_x$ and $NO_x$ below their emission reduction requirements, and as a result EU28 as a whole could be on track to overshoot the reduction requirements for these species by 2030. But for ammonia further efforts are needed in order to meet the 2030 commitments for many countries in Europe (IIASA, 2018). Cost-effective measures to further reduce ammonia emissions differ among various parts of Europe.
According to IIASA (2020) the damage cost estimate of 17.50 Euro per kg ammonia emitted is much higher than the average abatement costs. Also Giannakis et al. (2019) find that much more ambitious commitments for ammonia emission reductions could be applied by EU-28 countries with relatively minimal costs. According to Giannakis et al. (2019) low emission animal housing would be the least cost effective measure, but still a quarter of the costs of the avoided damage. However, this measure could be the most effective one to reduce emissions in winter, when $PM_{2.5}$ health related limits are the most likely to be
exceeded in Europe.

*Code availability.* The EMEP model version rv4.34 is available as open source code through https://doi.org/10.5281/zenodo.3647990(EMEP MSC-W, 2020).





*Data availability.* Model results are available upon request to first author

*Author contributions.* JEJ made the model calculations and wrote most of the paper. HF assisted in designing the model scenarios and in
writing the paper. TS calculated the exceedances of critical loads for eutrophication and contributed in the writing of the paper.

*Competing interests.* The authors declare that they have no conflict of interest.

*Acknowledgements.* Computer time for EMEP model runs was supported by the Research Council of Norway through the NOTUR project
EMEP (NN2890K) for CPU and the NorStore project European Monitoring and Evaluation Programme (NS9005K) for storage of data.

*Financial support.* This work has been partially funded by EMEP under the United Nations Economic Commission for Europe (UN ECE).





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

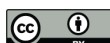



**Table 1.** Emissions (Em.) of $NO_x$ and ammonia and depositions (Dep.) of oxidised and reduced nitrogen and the ratio of reduced versus total (reduced + oxidised) deposition of nitrogen. Emissions and deposition are listed in 100 Mg of N. (Bosnia H. is Bosnia and Herzegovina and N. Macedonia is North Macedonia.

| | 2005 | | | | | 2017 | | | | | 2030 | | | | |
| | Emissions | | Depositions | | Red fr | Emissions | | Depositions | | Red fr | Emissions | | Depositions | | Red fr |
| Country | $NO_x$ | $NH_3$ | ox.N | red.N | | $NO_x$ | $NH_3$ | ox.N | red.N | | $NO_x$ | $NH_3$ | ox.N | red.N | |
| **28 EU countries** | | | | | | | | | | | | | | | |
| Austria | 724 | 516 | 631 | 589 | 48 | 441 | 569 | 423 | 587 | 58 | 224 | 454 | 261 | 487 | 65 |
| Belgium | 968 | 620 | 326 | 339 | 51 | 536 | 550 | 218 | 320 | 59 | 397 | 539 | 144 | 299 | 67 |
| Bulgaria | 581 | 425 | 552 | 444 | 45 | 313 | 407 | 400 | 451 | 53 | 245 | 374 | 310 | 408 | 57 |
| Croatia | 265 | 392 | 412 | 370 | 47 | 167 | 310 | 281 | 331 | 54 | 114 | 294 | 198 | 306 | 57 |
| Cyprus | 64 | 62 | 26 | 20 | 43 | 46 | 53 | 23 | 22 | 49 | 29 | 49 | 21 | 22 | 51 |
| Czechia | 840 | 636 | 669 | 587 | 47 | 496 | 552 | 462 | 555 | 55 | 303 | 496 | 283 | 465 | 62 |
| Denmark | 627 | 729 | 295 | 359 | 55 | 340 | 629 | 212 | 319 | 60 | 200 | 554 | 134 | 277 | 67 |
| Estonia | 128 | 84 | 168 | 108 | 39 | 100 | 84 | 133 | 104 | 44 | 89 | 84 | 98 | 95 | 49 |
| Finland | 633 | 307 | 628 | 351 | 36 | 396 | 256 | 505 | 347 | 41 | 335 | 246 | 361 | 281 | 44 |
| France | 4322 | 4980 | 2939 | 3423 | 54 | 2456 | 4994 | 1899 | 3393 | 64 | 1340 | 4333 | 1199 | 3054 | 72 |
| Germany | 4821 | 5267 | 3542 | 3943 | 53 | 3616 | 5544 | 2493 | 4010 | 62 | 1687 | 3740 | 1501 | 3039 | 67 |
| GB | 5408 | 2343 | 1315 | 1177 | 47 | 2718 | 2332 | 844 | 1148 | 58 | 1460 | 1968 | 518 | 1042 | 67 |
| Greece | 1430 | 533 | 657 | 337 | 34 | 776 | 459 | 443 | 336 | 43 | 644 | 479 | 402 | 336 | 46 |
| Hungary | 536 | 709 | 595 | 560 | 48 | 362 | 722 | 421 | 550 | 57 | 114 | 482 | 274 | 442 | 62 |
| Ireland | 517 | 933 | 146 | 392 | 73 | 335 | 976 | 107 | 414 | 79 | 769 | 887 | 67 | 379 | 85 |
| Italy | 3896 | 3515 | 2258 | 2215 | 50 | 2158 | 3164 | 1467 | 2008 | 58 | 1364 | 2953 | 1081 | 1923 | 64 |
| Latvia | 128 | 123 | 267 | 202 | 43 | 113 | 136 | 219 | 209 | 49 | 85 | 122 | 152 | 179 | 54 |
| Lithuania | 189 | 257 | 311 | 300 | 49 | 161 | 243 | 254 | 297 | 54 | 93 | 231 | 172 | 266 | 61 |
| Luxembourg | 167 | 48 | 26 | 26 | 50 | 55 | 48 | 15 | 26 | 63 | 28 | 38 | 9 | 22 | 71 |
| Malta | 30 | 12 | 2 | 1 | 33 | 15 | 9 | 2 | 2 | 50 | 6 | 9 | 2 | 1 | 33 |
| Netherlands | 1242 | 1274 | 410 | 554 | 57 | 767 | 1088 | 308 | 545 | 64 | 484 | 1006 | 191 | 459 | 71 |
| Poland | 2645 | 2671 | 2406 | 2303 | 49 | 2447 | 2533 | 1945 | 2240 | 54 | 1614 | 2217 | 1237 | 1908 | 61 |
| Portugal | 816 | 516 | 306 | 242 | 44 | 484 | 474 | 216 | 233 | 52 | 302 | 439 | 146 | 219 | 60 |
| Romania | 992 | 1697 | 1192 | 1265 | 51 | 706 | 1353 | 888 | 1164 | 57 | 397 | 1273 | 640 | 1062 | 62 |
| Slovakia | 313 | 312 | 352 | 303 | 46 | 201 | 219 | 249 | 268 | 51 | 157 | 219 | 167 | 236 | 62 |
| Slovenia | 167 | 166 | 173 | 136 | 44 | 107 | 153 | 117 | 146 | 56 | 16 | 119 | 69 | 112 | 62 |
| Spain | 4151 | 4173 | 1866 | 1699 | 48 | 2249 | 4267 | 1167 | 2044 | 64 | 416 | 3388 | 667 | 1570 | 70 |
| Sweden | 560 | 477 | 931 | 610 | 40 | 377 | 439 | 746 | 680 | 48 | 190 | 396 | 478 | 491 | 51 |
| **EU28** | **32848** | **29440** | **23401** | **22855** | **49** | **22938** | **32562** | **16457** | **22648** | **58** | **12131** | **23882** | **10782** | **19380** | **64** |
| **non EU countries** | | | | | | | | | | | | | | | |
| Switzerland | 280 | 497 | 257 | 343 | 57 | 186 | 454 | 171 | 373 | 69 | 189 | 446 | 123 | 302 | 71 |
| Iceland | 88 | 26 | 39 | 26 | 40 | 70 | 43 | 43 | 37 | 46 | 17 | 33 | 28 | 25 | 47 |
| Norway | 660 | 179 | 486 | 247 | 34 | 496 | 275 | 408 | 307 | 43 | 281 | 192 | 274 | 214 | 44 |
| Albania | 76 | 138 | 123 | 86 | 41 | 76 | 199 | 92 | 124 | 57 | 72 | 197 | 78 | 103 | 57 |
| Turkey | 2042 | 2355 | 2382 | 1626 | 41 | 2389 | 6092 | 2139 | 3822 | 64 | 2421 | 3992 | 2074 | 2555 | 55 |
| Bosnia H. | 100 | 130 | 286 | 225 | 44 | 94 | 175 | 198 | 240 | 55 | 103 | 205 | 147 | 215 | 59 |
| N. Macedonia | 113 | 65 | 121 | 69 | 36 | 73 | 84 | 80 | 76 | 49 | 73 | 58 | 70 | 65 | 48 |
| Serbia | 508 | 467 | 502 | 389 | 44 | 450 | 535 | 359 | 432 | 55 | 213 | 250 | 198 | 306 | 61 |
| Montenegro | 24 | 19 | 55 | 39 | 41 | 43 | 17 | 42 | 35 | 45 | 13 | 16 | 33 | 37 | 53 |





**Table 2.** Model runs performed. Base denotes model runs with all emissions for the years 2005, 2017, and 2030. For 2005 emissions in EU28 are based on EMEP 2005 official emissions. Remaining land based emissions from ECLIPSEv6a. For 2017 all emissions are as reported in EMEP Status Report 1/2020 (2020) appendix B. For 2030 EU28 emissions are scaled according to the NEC2030 obligations based on the 2005 emissions. Remaining land based emissions from Eclipse v6a. Emissions and model runs are also described in section 3. The additional model sensitivity runs reducing the emissions in steps of 10% are also listed.

| Year | Base | percentage emission reductions | | | | | | |
| | | NH$_3$ | | | | | NO$_x$ | SO$_x$ |
| | | -10% | -20% | -30% | -40% | -50% | -10% | -10% |
| 2005 | ✓ | ✓ | ✓ | | | | ✓ | ✓ |
| 2017 | ✓ | | | | | | | |
| 2030 | ✓ | ✓ | ✓ | ✓ | ✓ | ✓ | ✓ | ✓ |





**Table 3.** Exceedance of CL for Eutrophication (CLex eut.) by deposition (Dep.) of ammonia and reduced nitrogen. Exceedance are expressed as share [%] of the receptor area.

| Country | Eco Area 1000 km2 | CLex eut.[%] 2005 | 2017 | 2030 |
|---|---|---|---|---|
| **28 EU countries** | | | | |
| Austria | 50.4 | 73.5 | 56.6 | 29.7 |
| Belgium | 9.2 | 100.0 | 100.0 | 99.5 |
| Bulgaria | 48.9 | 99.9 | 99.5 | 98.1 |
| Croatia | 32.7 | 99.0 | 94.3 | 84.6 |
| Cyprus | 1.6 | 100.0 | 100.0 | 100.0 |
| Czechia | 6.4 | 100.0 | 99.6 | 86.1 |
| Denmark | 5.1 | 100.0 | 100.0 | 97.9 |
| Estonia | 18.9 | 80.6 | 75.7 | 42.0 |
| Finland | 40.9 | 10.2 | 7.8 | 0.9 |
| France | 176.3 | 78.0 | 62.0 | 46.1 |
| Germany | 101.3 | 83.6 | 77.0 | 64.7 |
| Greece | 64.4 | 98.0 | 94.1 | 93.2 |
| Hungary | 22.8 | 98.3 | 96.5 | 77.3 |
| Ireland | 12.8 | 25.0 | 22.2 | 14.8 |
| Italy | 105.7 | 77.6 | 60.0 | 48.7 |
| Latvia | 30.7 | 96.9 | 95.1 | 78.2 |
| Lithuania | 19.1 | 99.7 | 99.2 | 96.3 |
| Luxembourg | 1.0 | 100.0 | 100.0 | 100.0 |
| Malta | <1 | 94.8 | 94.8 | 94.8 |
| Netherlands | 0.4 | 85.2 | 78.3 | 70.0 |
| Poland | 91.2 | 78.2 | 70.8 | 48.5 |
| Portugal | 33.9 | 98.5 | 93.0 | 85.4 |
| Romania | 95.0 | 96.5 | 93.8 | 82.7 |
| Slovakia | 21.8 | 99.8 | 98.3 | 92.5 |
| Slovenia | 10.5 | 100.0 | 99.8 | 87.8 |
| Spain | 195.8 | 99.7 | 98.2 | 95.2 |
| Sweden | 58.6 | 14.3 | 12.8 | 8.4 |
| United Kingdom | 54.3 | 32.5 | 18.0 | 9.2 |
| **EU28** | 1,309.7 | 80.4 | 73.3 | 62.5 |
| **non EU countries** | | | | |
| Albania | 17.4 | 88.3 | 87.0 | 81.8 |
| Belarus | 55.0 | 100.0 | 100.0 | 99.3 |
| Bosnia & Herzegovina | 29.7 | 87.3 | 80.0 | 81.8 |
| Kosovo | 4.0 | 75.7 | 66.6 | 55.3 |
| Liechtenstein | <1 | 100.0 | 99.6 | 100.0 |
| North Macedonia | 13.1 | 79.9 | 69.9 | 61.9 |
| Moldova, Rep. of | 3.4 | 88.0 | 87.7 | 73.3 |
| Montenegro | 7.0 | 77.8 | 67.1 | 60.7 |
| Norway | 302.6 | 11.8 | 10.1 | 3.8 |
| Russia | 607.2 | 56.8 | 54.0 | 39.6 |
| Serbia | 28.9 | 94.8 | 89.8 | 75.4 |
| Switzerland | 7.5 | 82.1 | 74.3 | 55.5 |
| Ukraine | 91.2 | 99.8 | 99.8 | 98.6 |
| **Europe** | 2,476.8 | 67.7 | 62.8 | 52.3 |





**Table 4.** First column listing annual and seasonal concentrations of $PM_{2.5}$ as an average for the EU28 countries. Column 2 - 5 are listing the EU28 average reductions calculated in steps of 10% reductions in ammonia emissions. For $PM_{2.5}$ the reductions in $ngNm^{-3}$ per Gg of reduction of ammonia emissions are shown In brackets. The Corresponding effects of 10 and 20% reductions of ammonia emissions in 2005 are also shown. The effects of 10% reductions of $SO_x$ and $NO_x$ emissions in 2030 are also listed. The reductions in $PM_{2.5}$ in $ngNm^{-3}$ per Gg emitted are given in brackets with ammonia counted as $NH_3$ with molecular weight 17), $NO_x$ counted as $NO_2$ with molecular weight 46 and $SO_x$ counted as $SO_2$ with molecular weight 64.

| Season | Conc. $\mu gm^{-3}$ | 10% – Base $\mu gm^{-3}$ | 20% – 10% $\mu gm^{-3}$ | 30% – 20% $\mu gm^{-3}$ | 40% - 30% $\mu gm^{-3}$ | 50% – 40% $\mu gm^{-3}$ |
|---|---|---|---|---|---|---|
| **$PM_{2.5}$ 2030, $NH_3$ reductions** | | | | | | |
| Annual | 4.45 | -0.066 (0.23) | -0.074 (0.26) | -0.082 (0.28) | -0.092 (0.32) | -0.103 (0.35) |
| Winter | 5.31 | -0.126 (0.44) | -0.142 (0.49) | -0.160 (0.55) | -0.180 (0.62) | -0.202 (0.70) |
| Spring | 3.90 | -0.055 (0.19) | -0.061 (0.21) | -0.068 (0.23) | -0.77 (0.26) | -0.087 (0.30) |
| Summer | 4.23 | -0.015 (0.05) | -0.016 (0.06) | -0.016 (0.06) | -0.017 (0.06) | -0.018 (0.07) |
| Autumn | 4.39 | -0.069 (0.24) | -0.076 (0.26) | -0.086 (0.30) | -0.096 (0.33) | -0.108 (0.37) |
| **$PM_{2.5}$ 2030, $NO_x$ reductions** | | | | | | |
| Annual | 4.45 | -0.094 (0.27) | | | | |
| Winter | 5.31 | -0.112 (0.32) | | | | |
| Spring | 3.90 | -0.105 (0.30) | | | | |
| Summer | 4.23 | -0.051 (0.15) | | | | |
| Autumn | 4.39 | -0.106 (0.38) | | | | |
| **$PM_{2.5}$ 2030,$SO_x$ reductions** | | | | | | |
| Annual | 4.45 | -0.085 (0.58) | | | | |
| Winter | 5.31 | -0.101 (0.69) | | | | |
| Spring | 3.90 | -0.082 (0.56) | | | | |
| Summer | 4.23 | -0.071 (0.48) | | | | |
| Autumn | 4.39 | -0.086 (0.59) | | | | |
| **$PM_{2.5}$ 2005, $NH_3$ reductions** | | | | | | |
| Annual | 8.87 | -0.22 (0.61) | -0.23 (0.64) | | | |
| **$PM_{2.5}$ 2005, NOx reductions** | | | | | | |
| Annual | 8.87 | -0.16 (0.15) | | | | |
| **$PM_{2.5}$ 2005, SOx reductions** | | | | | | |
| Annual | 8.87 | -0.24 (0.37) | | | | |





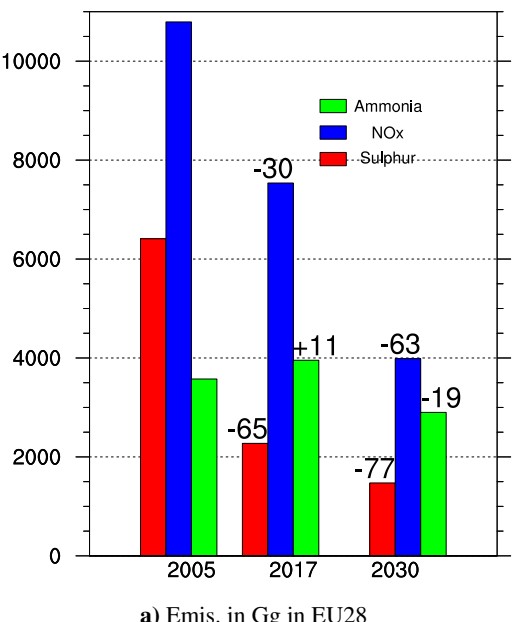

**a)** Emis. in Gg in EU28

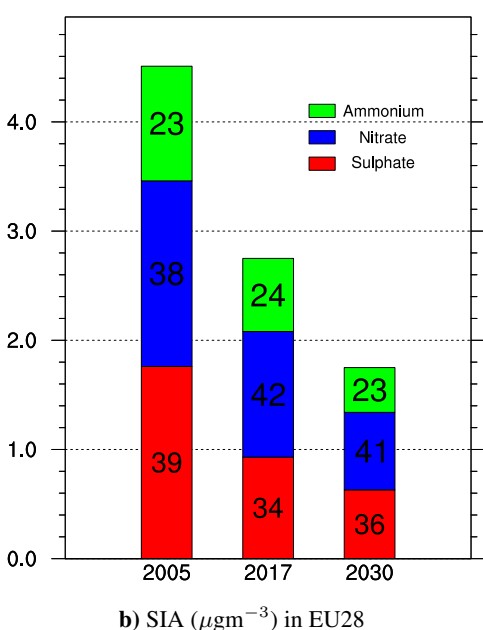

**b)** SIA ($\mu$gm$^{-3}$) in EU28

**Figure 1.** a: Emissions of SO$_x$ as SO$_2$, NO$_x$ as NO$_2$, and ammonia as NH$_3$ summed up for the EU28 countries. Percentage change in emissions from 2005 are given above the bars. b: Concentrations of SIA ($\mu$gm$^{-3}$) split into sulphate, nitrate, and ammonium and averaged over the EU28 countries. Percentage contributions to SIA are printed inside the bars.





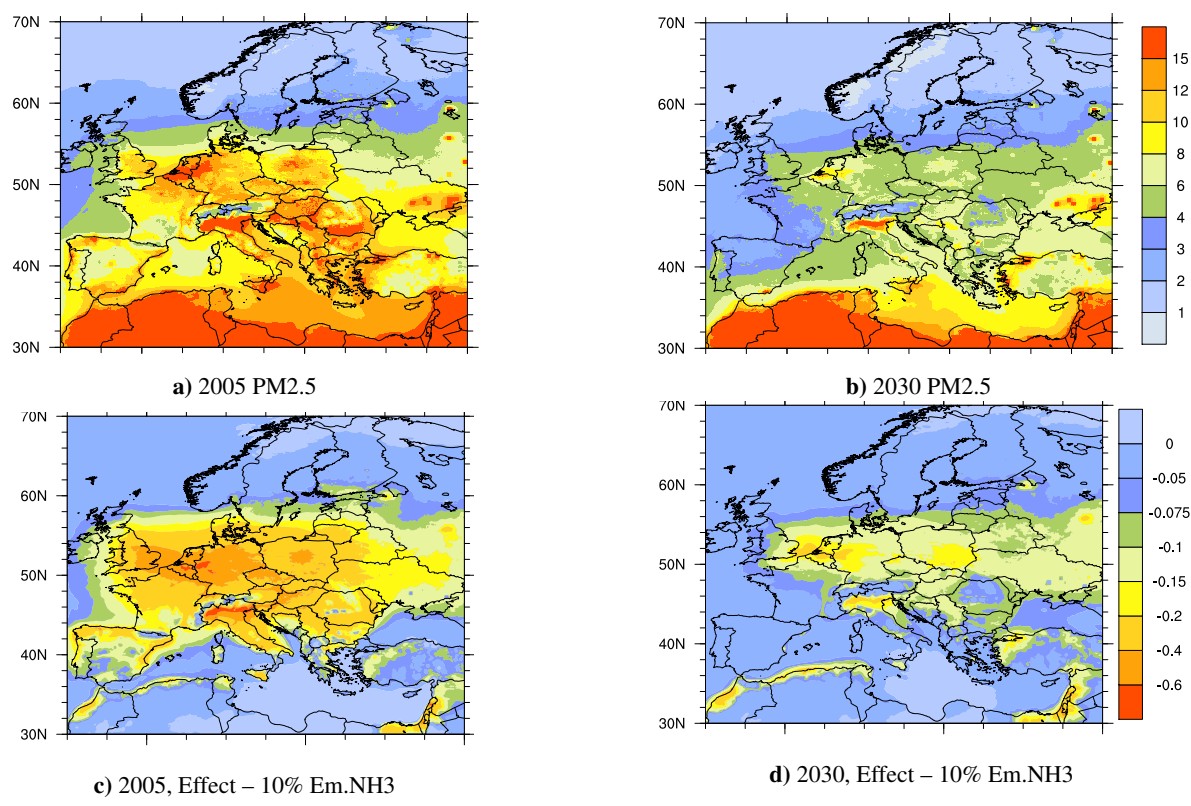

**Figure 2.** $PM_{2.5}$ in 2005 (a), and in 2030 (b). Effects of 10% further reductions in $NH_3$ emissions in 2005 (c) and in 2030 (d) [$\mu gm^{-3}$].





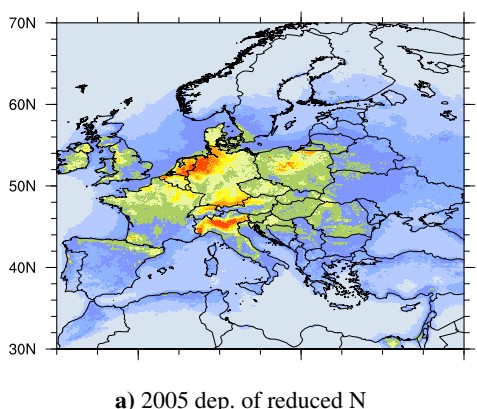

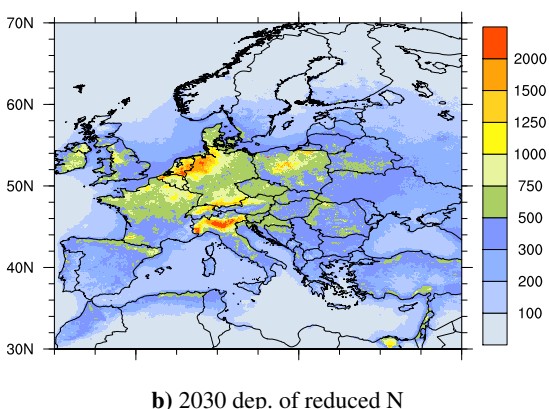

**a)** 2005 dep. of reduced N

**b)** 2030 dep. of reduced N

**Figure 3.** Deposition of reduced N in 2005 (a) and in 2030 (b) $[mg(N)m^{-2}]$.





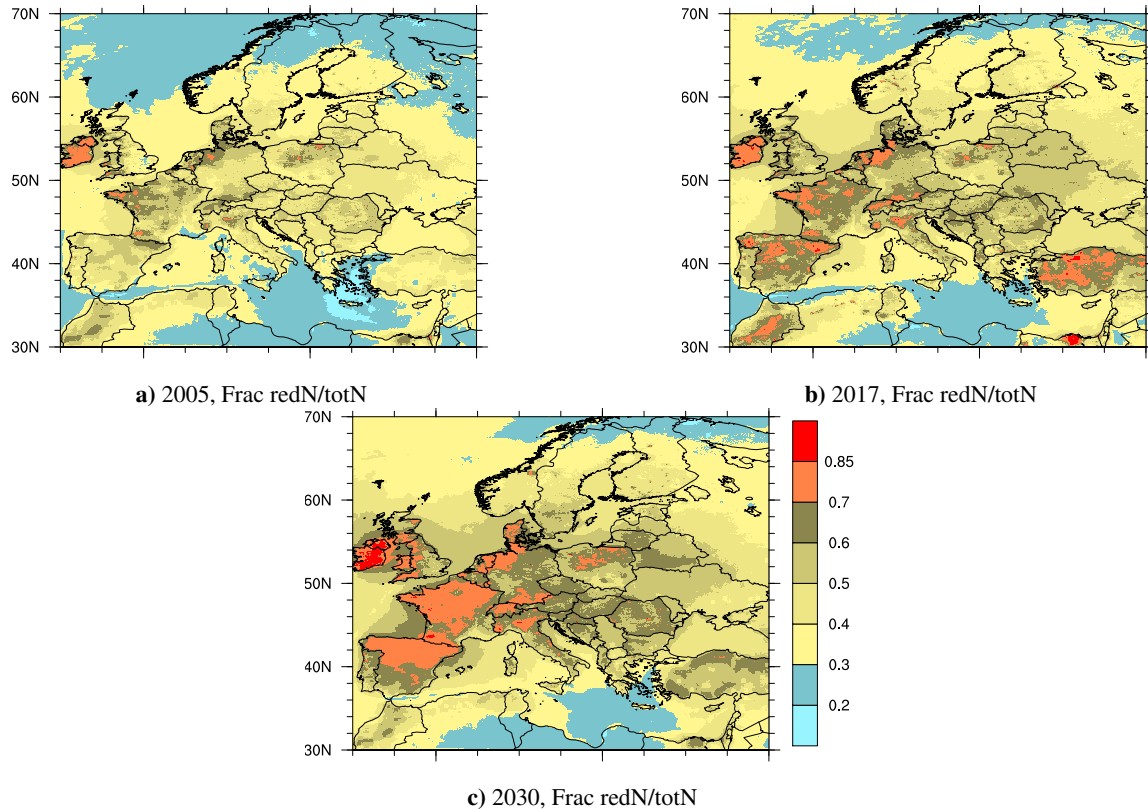

**a)** 2005, Frac redN/totN

**b)** 2017, Frac redN/totN

**c)** 2030, Frac redN/totN

**Figure 4.** Fraction of reduced N deposition relative to total N (reduced pluss oxidised nitrogen) deposition, calculated with year 2005, 2017 and 2030 emissions.

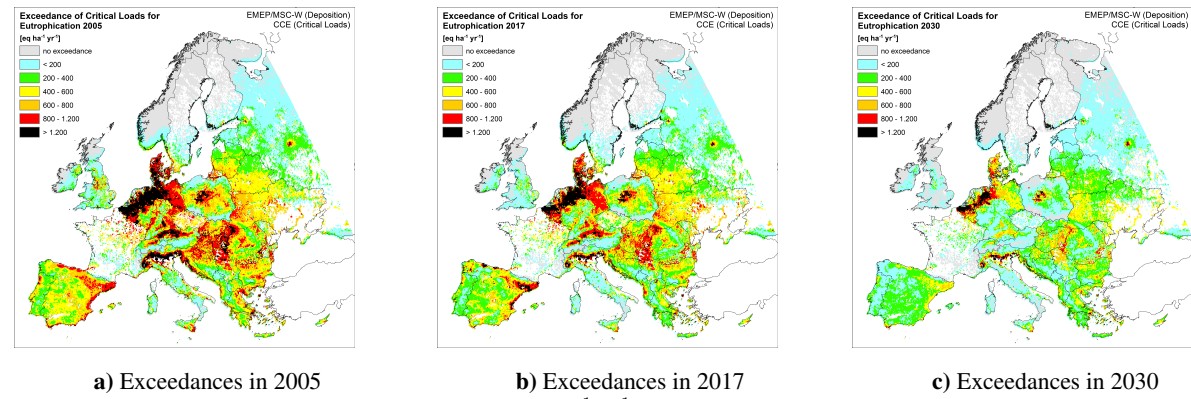

**a)** Exceedances in 2005      **b)** Exceedances in 2017      **c)** Exceedances in 2030

**Figure 5.** Calculated exceedanses of CL for eutrophication (eq ha$^{-1}$y$^{-1}$) in 2005, 2017, and 2030.



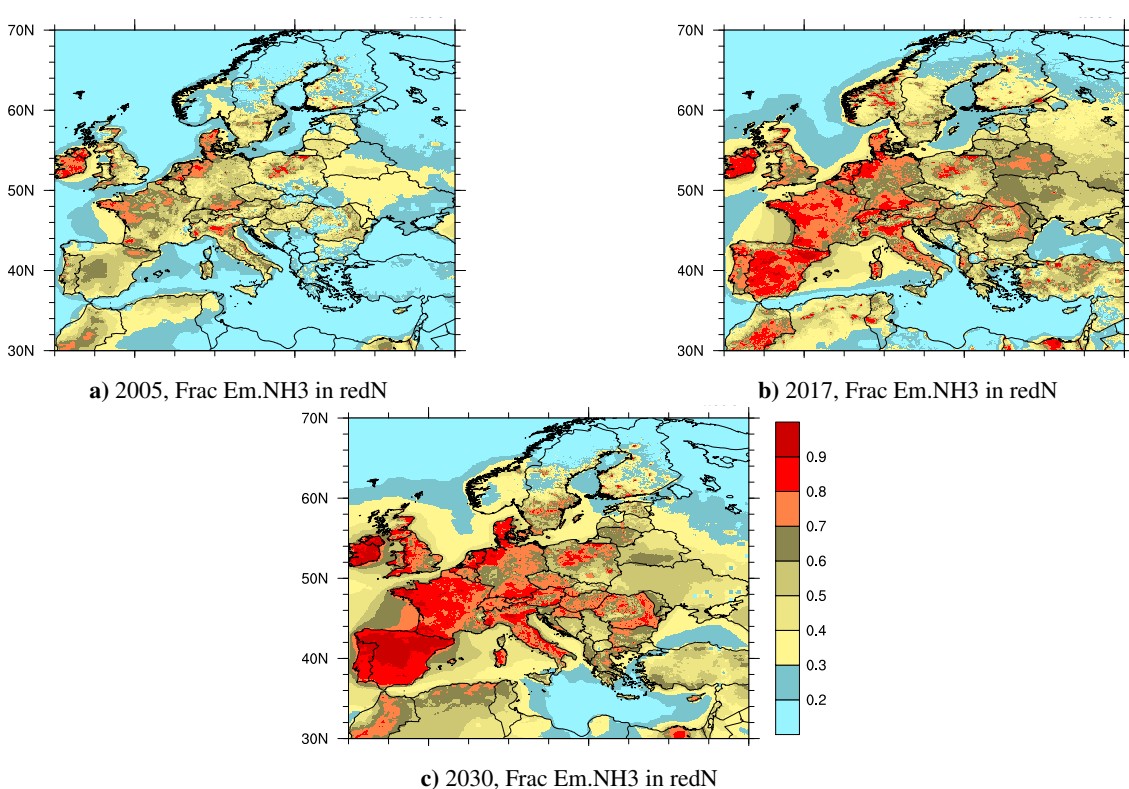

**a)** 2005, Frac Em.NH3 in redN

**b)** 2017, Frac Em.NH3 in redN

**c)** 2030, Frac Em.NH3 in redN

**Figure 6.** Fraction of NH$_3$ in reduced N (ammonia + ammonium) in 2005, 2017, and 2030.

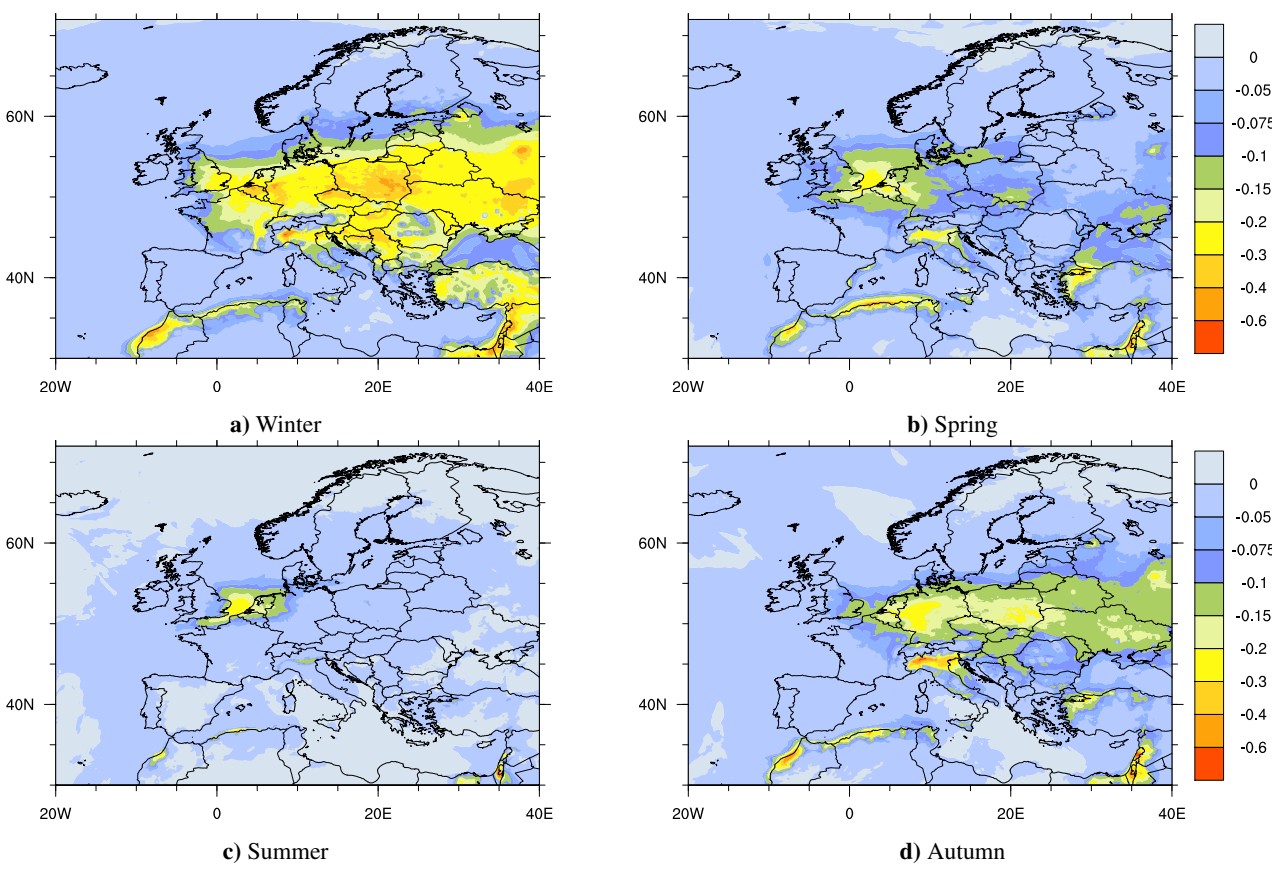

**Figure 7.** Effects of a 10% decrease in 2030 ammonia emissions on PM$_{2.5}$ [$\mu$gm$^{-3}$] split by season. Winter: December, January, February. Spring: March, April, May. Summer: June, July, August. Autumn: September, October, November.

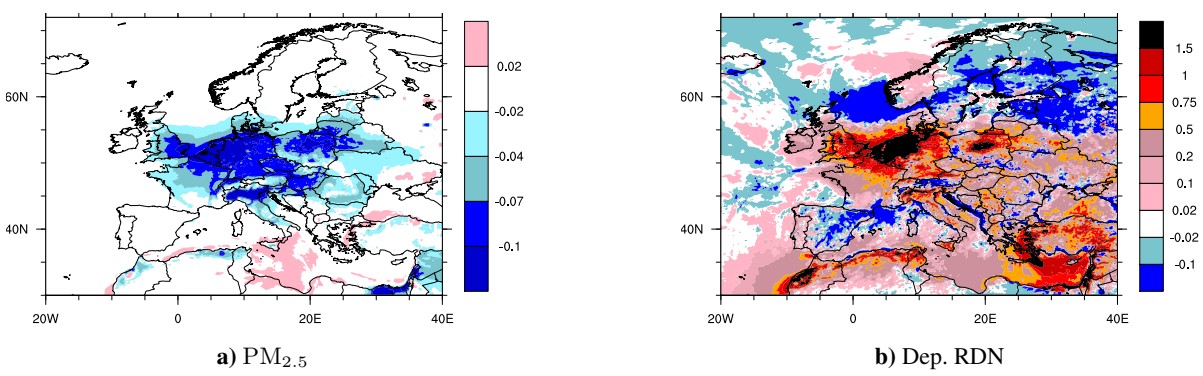

**Figure 8.** Difference between additional 50% − 40% reduction and additional 10% − Base in ammonia emissions on top of NEC 2030 for PM$_{2.5}$ concentrations [$\mu$gm$^{-3}$] (a), and deposition of reduced nitrogen [mg(N)m$^{-2}$] (b).





## Appendix A

In this appendix we show scatter-plots of EMEP model results versus measurements for some key species for the years 2005 and 2017. Several of the 2017 scatter-plots have already been published in Gauss et al. (2019), but are shown here in order to provide a direct comparison to the year 2005 results. For year 2017 the comparison to measurements differ slightly from the results shown in Gauss et al. (2019) as several new measurements have been made available since the publication of the above-mentioned report. For both years the scatter plots are shown with all measurements included. However, the selection of available measurements differs between the two years. In general there are more measurements available in 2017 compared to 2005. In Table A1 statistics for the scatter plots are listed for the same species as in the scatter plots, limiting the sites to those having measurements for both 2005 and 2017. The number of common measurements for both years are given in the table. In particular for ammonia in air and $HNO_3$ in air there are very few sites with measurements for both years.

**Table A1.** Year 2005 and year 2017 statistics for scatter plots with common measurements sites for both years. Obs is observed annual meal for the sites include. Bias is model bias and Corr. is correlation between measurements and the model calculations.

| Species | Nr. obs. | 2005 | | | 2017 | | |
|---|---|---|---|---|---|---|---|
| | | obs. | Bias | Corr. | obs. | Bias | Corr. |
| $PM_{2.5}$ | 14 | 11.34 | -14% | 0.57 | 7.75 | -18% | 0.84 |
| Ammonium in air | 13 | 0.94 | -19% | 0.94 | 0.57 | -18% | 0.89 |
| $NO_3^-$ in air | 13 | 1.68 | 5% | 0.71 | 1.44 | -5% | 0.80 |
| Ammonia in air | 7 | 0.76 | 26% | 0.78 | 0.73 | 40% | 0.90 |
| $HNO_3$ in air | 6 | 0.16 | -12% | 0.84 | 0.14 | -33% | 0.91 |





**a)** Ammonia ($NH_3$) in air ($\mu$g(N) m$^{-3}$) in 2005

**b)** Ammonia ($NH_3$) in air ($\mu$g(N) m$^{-3}$) in 2017

**c)** Ammonium ($NH_4^+$) in air ($\mu$g(N) m$^{-3}$) in 2005

**d)** Ammonium ($NH_4^+$) in air ($\mu$g(N) m$^{-3}$) in 2017

**Figure A1.** Scatter plots of modelled versus observed concentrations of ammonia (top) and ammonium (bottom) in air for the years 2005 (left) and 2017 (right). Statistics with limited set of common measurement sites are given in Table A1. Site positions listed in Tables A2 and A3. Measurements downloaded from http://ebas.nilu.no/default.aspx.





**Figure A2.** Scatter plots of modelled versus observed concentrations of nitric acid (top) and nitrate (bottom) in air for the years 2005 (left) and 2017 (right). Statistics with limited set of common measurement sites are given in Table A1. Site positions listed in Tables A2 and A3. Measurements downloaded from http://ebas.nilu.no/default.aspx.





**a)** Wet deposition of reduced nitrogen (mg(N)m$^{-2}$ in 2005

**b)** wet deposition of reduced nitrogen (mg(N)m$^{-2}$ in 2017

**c)** Wet deposition of oxidised nitrogen (mg(N)m$^{-2}$ in 2005

**c)** Wet deposition of oxidised nitrogen (mg(N)m$^{-2}$ in 2017

**Figure A3.** Scatter plots of modelled versus observed wet depositions of reduced nitrogen (top) and oxidised nitrogen (bottom) for the years 2005 (left) and 2017 (right). Statistics with limited set of common measurement sites are given in Table A1. Site positions listed in Tables A2 and A3. Measurements downloaded from http://ebas.nilu.no/default.aspx.





**a)** Sulphate in air ($\mu$g(S) m$^{-3}$) in 2005

**d)** Sulphate in air ($\mu$g(S) m$^{-3}$) in 2017

**c)** Wet deposition of sulphate (mg(S)m$^{-2}$ in 2005

**c)** Wet deposition of sulphate (mg(S)m$^{-2}$ in 2017

**Figure A4.** Scatter plots of modelled versus observed concentrations of sulphate in air (top) and wet deposition of sulphate (bottom) for the years 2005 (left) and 2017 (right). Statistics with limited set of common measurement sites are given in Table A1. Site positions listed in Tables A2 and A3. Measurements downloaded from http://ebas.nilu.no/default.aspx.





**Table A2.** List of sites included in scatter plots in Figures A1 to A4. Wet depositions when included for all three species (reduced nitrogen, oxidised nitrogen, and oxidised sulphur) marked with √. Only reduced and oxidised nitrogen marked with ∇, only oxidised N and S marked with △, only reduced N and oxidised S marked with ●, and only oxidised sulphur marked with ◇.

| Site | Lat. | Lon. | NH$_3$ | | NH$_4^-$ | | HNO$_3$ | | NO$_3^-$ | | Wdep | |
|------|------|------|------|------|------|------|------|------|------|------|------|------|
| | | | 2005 | 2017 | 2005 | 2017 | 2005 | 2017 | 2005 | 2017 | 2005 | 2017 |
| IS0002 | 64.08 | -21.02 | | | | | | | | | ◇ | △ |
| NO0001 | 58.38 | 8.25 | √ | | √ | | √ | | √ | | √ | √ |
| NO0002 | 58.38 | 8.22 | | √ | | √ | | √ | | √ | | |
| NO0008 | 58.82 | 6.72 | | | | | | | | | √ | |
| NO0015 | 65.83 | 13.92 | √ | √ | √ | √ | √ | √ | √ | √ | √ | √ |
| NO0042 | 78.91 | 11.89 | √ | | √ | √ | √ | | √ | | | |
| NO0055 | 64.47 | 25.22 | √ | | √ | | √ | | √ | | √ | |
| NO0056 | 60.37 | 11.08 | √ | √ | √ | √ | √ | √ | √ | √ | √ | |
| SE0005 | 63.85 | 15.33 | | √ | | √ | | √ | | √ | | √ |
| SE0012 | 58.80 | 17.38 | | √ | | √ | | √ | | √ | | |
| SE0014 | 57.39 | 11.91 | | √ | | √ | | √ | | √ | √ | √ |
| SE0020 | 56.04 | 13.15 | | √ | | √ | | √ | | √ | | |
| DK0003 | 56.35 | 9.60 | √ | √ | √ | √ | | | | | | |
| DK0005 | 54.75 | 10.74 | √ | | √ | | | | | | | |
| DK0008 | 56.70 | 11.52 | √ | √ | √ | √ | | | | | | |
| DK0012 | 55.69 | 12.09 | | √ | | √ | | | | | | |
| DK0031 | 56.30 | 8.43 | √ | | √ | | | | | | | |
| FI0009 | 59.78 | 21.38 | | √ | √ | √ | | √ | | √ | | |
| FI0017 | 60.52 | 27.69 | | | √ | | | | | | | |
| FI0018 | 60.53 | 27.67 | | √ | | √ | | √ | | √ | | |
| FI0036 | 68.00 | 24.24 | | √ | √ | √ | | √ | | √ | | |
| EE0009 | 59.50 | 25.90 | | | | | | | | √ | ∇ | √ |
| LV0010 | 56.16 | 21.17 | | √ | | √ | | √ | √ | √ | √ | |
| LV0016 | 57.14 | 25.91 | | | √ | | | | √ | | √ | |
| LT0015 | 55.37 | 21.03 | | | | √ | | | | √ | √ | √ |
| RU0001 | 68.93 | 28.85 | | | √ | | | | √ | | | √ |
| RU0013 | 64.70 | 43.40 | | | | | | | | | √ | √ |
| RU0018 | 61.00 | 28.97 | | | √ | √ | | | √ | √ | √ | √ |
| RU0020 | 56.53 | 32.94 | | | | | | | √ | √ | √ | √ |
| BY0004 | 52.23 | 23.43 | | | | | | | | | √ | √ |
| IE0001 | 51.94 | -10.24 | | | | | | | | | √ | √ |
| IE0005 | 52.87 | -6.92 | | | √ | √ | | | √ | √ | √ | √ |
| IE0006 | 55.38 | -7.34 | | | √ | √ | | | √ | √ | | √ |
| IE0008 | 52.18 | -6.36 | | | √ | √ | | | √ | √ | | |
| IE0009 | 52.30 | -6.51 | | | | | | | | | | √ |
| GB0002 | 55.31 | -3.20 | | | | | | | | | √ | |
| GB0048 | 55.79 | -3.24 | | | | | | | | | | √ |
| GB1055 | 51.15 | -1.44 | | | | | | | | | | √ |
| NL0009 | 68.00 | 24.24 | | | √ | | | | √ | | √ | √ |
| NL0010 | 51.54 | 5.85 | √ | | √ | | | | √ | | | |
| NL0091 | 68.00 | 24.24 | | | √ | | | | √ | | | √ |
| DE0001 | 54.93 | 8.31 | √ | | √ | | √ | | √ | | | |
| DE0002 | 52.80 | 10.76 | √ | √ | √ | | √ | √ | √ | √ | | √ |
| DE0007 | 53.17 | 13.03 | √ | √ | √ | | √ | √ | √ | √ | | √ |
| DE0009 | 54.44 | 12.72 | √ | | | | √ | | √ | | | |





**Table A3.** See caption, Table A3

| Site | Lat. | Lon. | NH₃ | | NH₄⁻ | | HNO₃ | | NO₃⁻ | | Wdep | |
|---|---|---|---|---|---|---|---|---|---|---|---|---|
| | | | 2005 | 2017 | 2005 | 2017 | 2005 | 2017 | 2005 | 2017 | 2005 | 2017 |
| PL0002 | 51.81 | 21.97 | | | ✓ | ✓ | | | ✓ | ✓ | ✓ | ✓ |
| PL0004 | 54.75 | 17.53 | | | ✓ | ✓ | | | ✓ | ✓ | ✓ | ✓ |
| PL0005 | 54.15 | 22.07 | | ✓ | | ✓ | | ✓ | | ✓ | ✓ | ✓ |
| CZ0003 | 49.47 | 15.08 | | | | | | | | | ✓ | ✓ |
| SK0004 | 49.15 | 20.28 | | | | | | | | ✓ | ✓ | |
| SK0005 | 49.37 | 19.68 | | | | | ✓ | | ✓ | | | |
| SK0006 | 49.05 | 22.67 | | ✓ | | ✓ | ✓ | ✓ | ✓ | ✓ | ✓ | ✓ |
| SK0007 | 47.96 | 17.86 | | | | | ✓ | | ✓ | | | |
| CH0002 | 46.81 | 6.94 | | | | | | | | | ✓ | |
| CH0005 | 47.07 | 8.46 | | | | | | | | | ✓ | |
| AT0002 | 47.77 | 16.77 | ✓ | | ✓ | | ✓ | | ✓ | | ✓ | |
| AT0005 | 46.48 | 12.97 | | | | | | | | | ✓ | |
| AT0048 | 47.84 | 14.44 | | | | | | | | | ✓ | |
| HU0002 | 46.97 | 19.58 | ✓ | ✓ | ✓ | ✓ | ✓ | ✓ | ✓ | ✓ | ✓ | ✓ |
| HU0003 | 46.91 | 16.32 | | ✓ | | ✓ | | ✓ | | ✓ | | ✓ |
| MD0013 | 46.49 | 28.28 | | ✓ | | ✓ | | ✓ | | ✓ | | ✓ |
| ES0001 | 42.32 | 3.32 | | | | | | | | | | ✓ |
| ES0006 | 39.88 | 4.32 | | | | | | | | | | ✓ |
| ES0007 | 37.24 | -3.53 | | | | | | | | | ✓ | ✓ |
| ES0008 | 43.44 | -4.85 | | | | | | | | | ✓ | ✓ |
| ES0009 | 41.27 | -3.14 | | | | | | | | | ✓ | ✓ |
| ES0011 | 38.47 | -6.92 | | | | | | | | | ✓ | ✓ |
| ES0012 | 39.08 | -1.10 | | | | | | | | | ✓ | ✓ |
| ES0013 | 41.24 | -5.90 | | | | | | | | | ✓ | ✓ |
| ES0014 | 41.39 | -0.73 | | | | | | | | | ✓ | ✓ |
| ES0015 | 39.52 | -4.35 | | | | | | | | | ✓ | • |
| ES0016 | 42.63 | -7.70 | | | | | | | | | ✓ | ✓ |
| ES0017 | 37.05 | -6.51 | | | | | | | | | | ✓ |
| FR0008 | 48.50 | -7.13 | | | | | | | | | ✓ | ✓ |
| FR0009 | 49.90 | 4.63 | | | | | | | | | ✓ | ✓ |
| FR0010 | 47.27 | 4.08 | | | | | | | | | ✓ | ✓ |
| FR0012 | 43.03 | -1.08 | | | | | | | | | ✓ | |
| FR0013 | 43.62 | 0.18 | | | | | | | | | ✓ | ✓ |
| FR0014 | 47.30 | 6.83 | | | | | | | | | ✓ | ✓ |
| FR0015 | 46.65 | -0.75 | | | | | | | | | ✓ | ✓ |
| FR0016 | 45.00 | 6.47 | | | | | | | | | ✓ | ✓ |
| FR0017 | 45.80 | 2.07 | | | | | | | | | ✓ | |
| FR0018 | 48.63 | -0.45 | | | | | | | | | | ✓ |
| IT0001 | 42.10 | 12.63 | ✓ | | ✓ | | ✓ | | ✓ | ✓ | ✓ | |
| IT0004 | 45.80 | 8.63 | | | | | | | | | ✓ | ✓ |
| SI0008 | 45.47 | 14.87 | | ✓ | | ✓ | | ✓ | | ✓ | ✓ | ✓ |
| ME0008 | 43.15 | 19.13 | | | | | | | | | | • |
| RS0005 | 43.40 | 21.95 | | | | | | | | | ✓ | |
| TR0001 | 40.00 | 33.00 | ✓ | | ✓ | | ✓ | | ✓ | | ✓ | |
| GE0001 | 41.76 | 42.83 | | ✓ | | | | ✓ | | ✓ | | |
| AM0001 | 40.38 | 44.26 | | ✓ | | ✓ | | ✓ | | ✓ | | |