# Peer review of "Modelling changes in secondary inorganic aerosol formation and nitrogen deposition in Europe from 2005 to 2030"

_Atmospheric Chemistry and Physics, 2021_

## Author Response (AR1)

**Comments to remarks from reviewer 1**

**General comments:**

Several papers have investigated the interplay between SOx/NOx and NH3 emission changes. Nenes et al. (Atmos. Chem. Phys., 20, 3249–3258, 2020 and Atmos. Chem. Phys., 21, 6023–6033, 2021) provide a very interesting modeling framework to evaluate when particulate matter and dry deposition of inorganic reactive nitrogen are sensitive to ammonia and nitrate availability using aerosol pH and liquid water content as drivers. I think the present manuscript would benefit from a relevant discussion and comparison to these findings.

We have included a discussion of the implications of the results from the Nenes et al. papers in section 5 (Discussion and and conclusions):

As discussed in Nenes et al. (2020.021), gas/aerosol partitioning of total reduced and oxidised nitrogen are affected by aerosol pH level and water content, so that low (high) pH is favourable for  $NH_4^+$  (NO3-) formation. The increase in the aerosol fraction in total reduced and oxidised nitrogen would lead to decreases in their dry deposition, and subsequently their residence times and transport distances. This effect has not been accounted for in the EMEP model, thus some limited local effects might have been missed in our model simulations. For instance, based on the Nenes et al. (2021) results, there may be additional  $NO_3^-$  formation in areas with low acidity, such as coastal or dusty regions. Potentially this may reduce the deposition of total nitrate near these local sources, somewhat enhancing the accumulation of particles. Furthermore, as future emissions of SOx and NOy are expected to decrease, the pH of the particles is likely to increase, potentially favouring NO3 formation, and thus decreasing dry deposition and increasing the transport distances of oxidised and total nitrogen in some regions. On the other hand, our results show that overall, the fraction of reduced nitrogen in the total nitrogen has been increasing, and this increase is expected to continue until 2030. Assuming that the deposition rates for total nitrogen are mostly driven by those of reduced nitrogen (following Nenes et al. 2021)), the local effects of NO3 formation bursts would probably not play a major role across the regions in different present and future chemical regimes. Therefore we believe that overall, the main conclusions presented in our paper remain valid.

Minor corrections are listed below:

Line 51: sulphate sulphate replaced by SO42-

Line 73: differs Corrected

**Line 100-102: could you provide an equation for this ?**

The calculation is based on an extensive set of input data and equations. A detailed description of the calculation of critical loads is described in Chapter 5 in the mapping manual, see new text.

**New text:**

The CL exceedances presented here were calculated using the current CL database, which is described in \cite{Hettelingh:2017} and stored by the current Coordination Centre for Effects (CCE) at the German Federal Environmental Agency. The calculation is based on an extensive set of input data and equations. A detailed description is included in the Mapping Manual of the ICP Modelling and Mapping. (\cite{CLRTAP2017}, Chapter 5). This dataset is also used, among other things, to support European assessments and negotiations on emission reductions \cite[]{Hettelingh2001,Reis2012,EEA2014}.

**Line 134: as ammonium is either...**

Changed as suggested.

Line 145: is deposited than is emitted Changed as suggested.

Line 212: is small Changed as suggested.

**Figure 1a: emission units should be Gg.y-1**

Changed as suggested.

Figure 3 caption units should be mg(N).m-2.y-1 Changed as suggested.

Table 2 – please clarify what meteorological data are used for each simulation year.We have added:

All model runs have been performed with 2017 meteorological conditions as described in Section 3.

In figure 5 the CL exceedances are given in eq ha-1 y-1 while in Figure 3 for the deposition fluxes the surface unit is m2 – could you make them uniform ?

Deposition fluxes and CL are addressing different user groups. The practice in the air pollution community has been to use  $mg(N)m^{-2} y^{-1}$ , whereas the effects community prefers eq ha-1 y-1. This practice is also followed in the annual EMEP reporting (https://emep.int/mscw/mscw\_publications.html) to the Convention on "Long-range Transboundary Air Pollution"

**Comments to remarks from reviewer 2**

**General comments:**

**There are inconsistencies in the use of different types of subscripts for NOx, SOx etc.**

We have changed the naming of the species to be more consistent (ammonium replaced by NH3, ammonium replaced by NH4+ etc).

**There are different types of quotes used in different parts of the manuscript.**

We are not sure what quotes the reviewer is referring to here. We have however made substantial changes in the manuscript that hopefully has changed this problem.

**The manuscript needs a thorough read and correction of grammar and syntax.**

We have corrected errors pointed out by both reviewers, and also errors we have detected ourselves.

This may not be important, but there is no EU28 anymore since the exodus of the UK from EU. What the authors mean by the term is clear now (in the year 2021), but might not be for future readers. I suggest that the term EU28 is defined before first use. It can then remain in the text as is.

Near the top of section 2 (Model description) we have added: (EU28 includes the current EU27 countries and United Kingdom)

UK was added to the EU definition as they were part of EU in both 2005 and 2017. As a result they are/were committed to the emission ceiling directive.

**Specific comments:**

**P2 L28: There is no Appendix B.**

This refers to appendix B in the EMEP report. This has been made more clear.

**P2 L44: dot (.) missing after sulphate.**

Dot added.

**P2 L51: sulphat -> sulphate (missing e)**

Changed to SO42-

**P3 L80: Either ``both these studies" or ``These two studies"** Changed to "These two studies ... "

**P3 L80: Provide some numbers to support your claim.**

We have added more information about the model performance in these two studies:

Out of the 14 models included in the study by Vivanco et al. (2018) the EMEP model was one of very few with low fractional biases compared to measurements for the wet depositions of reduced nitrogen (-0.01), oxidised nitrogen (-0.05), and  $SO_4^{2^-}$  (-0.11). For the trend studies presented in Theobald et al. (2019) the fractional bias for the years 1990 to 2010 was -0.18, -0.02, and 0.22 for the wet deposition of reduced nitrogen, oxidised nitrogen, and  $SO_4^{2^-}$  respectively, but the overall overestimation chem $SO_4^{2^-}$  was mainly caused by an overestimation in the first years of the period.

**P4 L87: Either ``...nitrogen can also have acidifying impacts in ecosystems...", or ``nitrogen can also cause acidification of ecosystems..."**

Changed to:

".... nitrogen can also have acidifying impacts in ecosystems ....."

**P4 L107: There is no appendix B.**

Appendix B refers to appendix B in the EMEP report. Changed to:

.... listed in in appendix~B in EMEP (2020).

**P4 L109-115: This needs a bit of discussion: how do the ECLIPSEv6a emissions compare to the EMEP emissions for the countries that they were replaced? Are there significant differences? How do you account for the discontinuity of emissions between the datasets? What do you use for emissions for 2017 for the other countries?**

For non EU28 countries there are no EMEP 2030 emission projections available, so we chose to use the 2005 and 2030 Eclipse emissions for these years. For 2017 we have used the model run used in the 2019 EMEP report, as this provides a link to the oficial EMEP reporting process. In this section we have included more information about the 2017 emissions.

The description of how EMEP and Eclipse emissions are used and combined is improved in the first parts of section 3.

**P5 L150: Provide number of the portion that leaves the model domain.** New formulation here:

The remaining 0.2-0.3 is either deposited at sea or in non-EU countries. About 15% of the  $NH_3$  emitted within the model domain is advected out of the model domain, but much of this is coming from non EU countries close to the eastern model boundaries.

Table 1 caption: The (Em.) and (Dep.) are not used in the table, hence no need to define them. Also, since N. Macedonia and Bosnia H. are defined, GB should also be defined. If by GB you mean Great Britain, you should probably change it to UK to be also consistent with Table 3.

We have removed the definitions of (Em.) and (Dep.) from Table 1. GB is changed to UK, defined as the "The United Kingdom of Great Britain and Northern Ireland" in the caption.